# The SHED Index: A Validation Study to Assess Sustainable HEalthy Diets in Portugal

**DOI:** 10.3390/nu15245071

**Published:** 2023-12-12

**Authors:** Margarida Liz Martins, Sigal Tepper, Bebiana Marques, Sandra Abreu

**Affiliations:** 1Coimbra Health School (ESTeSC), Polytechnic University of Coimbra, 3045-093 Coimbra, Portugal; 2GreenUPorto—Sustainable Agrifood Production Research Centre, 4200-465 Vairao, Portugal; 3Centre for the Research and Technology of Agro-Environmental and Biological Sciences (CITAB), 5000-801 Vila Real, Portugal; 4Department of Nutritional Sciences, Tel-Hai College, Upper Galilee 1220800, Israel; sigalt@post.bgu.ac.il; 5Faculty of Nutrition and Food Sciences, University of Porto, 4150-180 Porto, Portugal; bmarques2000@hotmail.com; 6Research Centre in Physical Activity, Health, and Leisure (CIAFEL), Faculty of Sport, University of Porto, 4200-450 Porto, Portugal; sandraabreu@utad.pt; 7School of Life Sciences and Environment, University of Trás-os-Montes and Alto Douro (UTAD), 5000-801 Vila Real, Portugal; 8Laboratory for Integrative and Translational Research in Population Health, 4050-600 Porto, Portugal

**Keywords:** diet indexes, food sustainability, healthy eating, sustainable nutrition, validity

## Abstract

This study aims to adapt and validate the Sustainable HEalthy Diet (SHED) Index for the Portuguese adult population, which was developed to assess sustainable and healthy eating patterns. Data were collected using a web-based questionnaire administered through interviews with 347 individuals aged between 18 and 65 years old. The SHED Index evaluates 30 items, allowing for the assessment and scoring of sustainable and healthy eating patterns. The higher the SHED Index score, the more sustainable and healthier the diet. A semi-quantitative food frequency questionnaire was used to assess the participants’ dietary intake. The criterion validity was examined by testing the relationship between the SHED Index score and adherence to the Mediterranean Diet. Reproducibility was assessed by determining agreement and reliability with test–retest. Construct validity was confirmed based on established criteria. A higher SHED Index score was associated with moderate to high adherence to the Mediterranean diet, while it was inversely related to the proportion of animal-sourced foods in the overall food intake (r = −0.281, *p* < 0.001). Good reliability and agreement were found for the SHED Index score. Our findings suggest that the SHED Index is a valid and reliable tool for assessing sustainable and healthy diets in the Portuguese adult population.

## 1. Introduction

Human activities threaten our planet, as resources are being used at a rate that exceeds the Earth’s capacity to regenerate them [1]. Unsustainable food practices, occurring at different stages of production, storage, transportation, and consumption, are a major cause of environmental degradation and the depletion of natural resources [2,3]. Moreover, the current dietary trends, coupled with projected population growth, will further intensify the burden of non-communicable diseases and exacerbate the effects of global warming, including greenhouse gas emissions, nitrogen and phosphorus pollution, biodiversity loss, and increased water and land use [4,5].

Climate change is also impacting food systems and disrupting the food supply chain, and it is expected to put millions of people at risk of hunger, malnutrition, and poverty [6]. Despite growing concerns about food security [7], approximately 931 million tonnes of food were wasted in 2019 [8], while nearly 60% of adults were affected by obesity or being overweight [9]. Moreover, globalization and busy urban lifestyles have led modern societies to undergo a “nutrition transition” towards what is often referred to as the “Western diet”. This diet is characterized by convenient, processed, and nutritionally poor food choices, which are associated with negative environmental impacts and health risks [10,11,12,13]. For these reasons, food has gained a central role in our societal push towards a sustainable future [14]. And even though the first indications of ozone recovery are appearing [15], continued combined efforts and commitment are necessary to prevent rapid planet and health deterioration. The EAT Lancet Commission states that a transformation to healthy diets from sustainable food systems is necessary to achieve the United Nations Sustainable Development Goals and the Paris Agreement. Therefore, it is crucial to make changes to our current unhealthy and unsustainable diets in order to improve health and minimize environmental impact [4].

The World Health Organization and the Food and Agriculture Organization define sustainable healthy diets as “dietary patterns that promote all dimensions of individuals’ health and wellbeing, have low environmental pressure and impact, are accessible, affordable, safe and equitable, and are culturally acceptable” [3]. However, there is still uncertainty regarding the measurable components that make up a sustainable and healthy diet [2]. Some authors have already gathered existing knowledge and developed new tools to quantitatively monitor sustainable and healthy diets, based on the characteristics and dietary habits of different countries’ populations [16,17,18,19,20]. For example, the World Index for Sustainability and Health (WISH) [19] and the Planetary Health Diet Index (PHDI) [16] are based on the EAT-Lancet diet recommendations [4]. The scores for these indices are obtained by considering only the intake of specific food groups (e.g., whole grains, vegetables, fruits, dairy products, red meat, fish, eggs, chicken and other poultry, legumes, nuts, oils, and added sugars) [16,19]. Similarly, the Sustainable Diet Index (SDI), developed for the NutriNet-Santé study cohort, includes environmental, nutritional, economic, and sociocultural indicators but does not consider food waste [18]. On the other hand, the Healthy and Sustainable Diet Index (HSDI) considers components within five categories: animal-based foods, seasonal fruits and vegetables, ultra-processed energy-dense nutrient-poor foods, packaged foods, and food waste. However, it does not consider the origin and acquisition of food products [17]. Additionally, the Malaysia Diet Quality Index has been developed and validated for university students. This index is based on the Malaysian Dietary Guidelines and Food Pyramid, and includes five indicators: rice, animal-based food, plant-based food, food waste and packaging [20].

In Portugal, the sustainability of food consumption has been previously measured using Ecological Footprint Accounting. This method calculates the renewable natural resources and ecological services required due to the population’s consumption activities and compares it to the available biocapacity [1]. The results showed that Portugal has the highest food footprint among Mediterranean countries. Nevertheless, this method has some limitations as it is widely used for ecological studies and jointly assesses the impact of multiple pressures that are usually evaluated independently [14], and some indicators are not considered, like food waste.

On the other hand, Tepper et al. [21] developed the Sustainable HEalthy Diet (SHED) Index and validated it in the Israeli population. This index serves as a tool for measuring both healthy dietary patterns and pro-sustainability behaviors, offering several advantages. Firstly, the score considers the nutritional, environmental, and sociocultural aspects of sustainable diets. Additionally, it provides a practical and evidence-based tool designed to be used by health care professionals, individuals, and for research purposes, such as intervention studies, with input from a multidisciplinary advisory panel. Lastly, it is significantly correlated with the Mediterranean Diet, which is not only a part of Portuguese gastronomy but also recognized as a sustainable dietary pattern.

Given the importance of measuring and scoring healthy and sustainable diets, it is surprising that no methodological proposal has been described for the Portuguese population. Therefore, the aim of this study is to adapt and validate the SHED Index for the Portuguese adult population. This adaptation will enable the assessment of sustainable and healthy eating practices in Portugal.

## 2. Materials and Methods

### 2.1. Ethics

This research was approved by the Ethics Committee of Polytechnic University of Coimbra (100_CEIPC/2022). The study was conducted in accordance with the World Medical Association’s Helsinki Declaration for Human Studies. Written informed consent was obtained from participants beforehand.

### 2.2. Study Population

This study included a convenience sample of individuals aged between 18 and 65 years old at the time of recruitment. Academic institutions, schools, and private institutions for social solidarity were contacted to participate and disseminate information about the study. After this, individuals were invited through verbal communication, such as personal announcements and word-of-mouth. Individuals were deemed ineligible if they were unable to speak, understand, and write Portuguese, or if they had dementia or another mental condition that made them incapable of completing questionnaires.

The sample size was determined to detect a correlation coefficient ≥ 0.30 at the 0.05 significance level, with two-sided tails and considering a power of 80%. The sample size required was 84; assuming a drop-out rate of 20%, we aimed to reach at least 101 individuals. 

For convenience, data were collected using an electronic data form (Google Forms^TM^, https://docs.google.com/forms, accessed on 1 October 2022 that was completed by the participants in collaboration with the investigator. The questionnaire was administered to 351 participants from diverse population groups in order to fulfil the principle of maximum diversity through the convenience sampling method. For the present analysis, we excluded 4 participants who did not complete the food frequency questionnaire. The final sample included 347 individuals. To assess the reliability of the questionnaire, a sample of 84 participants completed a test–retest within three months of filling out the initial questionnaire.

### 2.3. Data Collection and Procedures

Data collection was conducted from 16 October to 15 December 2022 by three trained researchers through a web-based questionnaire. Before collecting data, the data collectors underwent a one-day training session to standardize procedures and maximize efficiency. In order to assess the comprehensibility of the questionnaires and address feasibility concerns for the main study data collection, a pre-test study was conducted with a sample of 10 adult volunteers. The maximum time required to complete the questionnaire was used as a measure of comprehensibility.

#### 2.3.1. Sociodemographic and Lifestyle Information

Sociodemographic and lifestyle data included sex, age, educational level, employment status, marital status, area of residence (urbanization degree), household composition, and physical activity level during leisure time using the International Physical Activity Questionnaire (IPAQ). The IPAQ Short-Form (IPAQ-SF) has been previously validated and adapted for the Portuguese population. It consists of 7 items in open-ended questions about individuals’ physical activity in the past 7 days [22].

Self-reported height and body weight were also collected, and the body mass index was calculated using the formula: weight/height^2^ (kg/m^2^). Participants’ body mass index was categorized according to the World Health Organization’s body mass index cut-points into non-overweight (≤18.5 to 24.9 kg/m^2^), overweight (25.0 to 29.9 kg/m^2^), and obese (≥30.0 kg/m^2^) [23].

#### 2.3.2. Sustainable and Healthy Diet (SHED) Index

The SHED Index, developed by Tepper et al. [21], was used to assess and score sustainable and healthy eating patterns. Firstly, the tool was translated into Portuguese and adapted to fit Portuguese culture and dietary habits. This was followed by a back-translation into English to ensure vocabulary consistency. The tool encompassed various dimensions related to overall dietary consumption (i.e., animal and plant-based foods, sugar and salt, and sugary and salty food products); consumption of sweetened beverages, and both bottled and tap water; consumption of ultra-processed foods and ready meals (e.g., frozen and takeaway); purchase of organic and local food (from a local groceries, farmer’s markets, farms, self-production); and food waste and recycling practices. 

The questionnaire includes 30 items. Sustainable and healthy eating were recorded on a Likert scale of 1–4, ranging from “Almost never true” to “Almost always true” or “Never” to “Most of the time”. Data on the consumption of beverages and ready meals were recorded on a scale of six frequencies, from “Never” to “Daily”. In the last question, participants were asked to self-perceive their consumption of plant-based foods in relation to their overall diet and rank this consumption on a 0–100% scale.

According to the original methodology [21], the questionnaire data underwent several transformations to allow the determination of six different sub-scores, which combined the items from the same dimension. The sub-scores included Healthy Eating, Sustainable Eating, Water, Ready-Meals, Sodas, and Buy Fruits and Vegetables (BFV). The “Healthy Eating” sub-score includes 10 items, while the “Sustainable Eating” sub-score includes 7 items. A higher “Healthy Eating” score indicates a healthier diet, while a higher “Sustainable Eating” score indicates more sustainable eating behavior. Other sub-scores are calculated by assigning different weights to each frequency of consumption. The “Water” sub-score reflects the source of drinking water, with a higher score indicating frequent use of tap water, and a lower score indicating frequent use of bottled water. The “Ready Meals” sub-score includes items related to eating out or consuming frozen or refrigerated meals. A higher score indicates a greater consumption of home-cooked meals. The “Sodas” sub-score evaluates the frequency of consuming sugar and artificially sweetened beverages, with more weight given to sugar-sweetened beverages. A higher “Sodas” sub-score indicates a lower frequency of consumption. The last sub-score is called “Buy Fruits and Vegetables” (BFV), and a higher score indicates a greater preference for purchasing local fruits and vegetables over non-local ones. The total SHED Index score combines information from six sub-scores and the proportion of the diet that is plant-based. As with the original SHED Index [21], the standardized final score was normalized for a mean of 60 and a standard deviation of 10. The higher the score, the more sustainable and healthy the diet.

#### 2.3.3. Dietary Assessment

To assess the dietary intake of participants over the past 12 months, a semi-quantitative food frequency questionnaire (FFQ) was used. The FFQ, consisting of 86 items, was developed and validated for the Portuguese adult population [24,25]. Participants were asked to indicate the frequency of consumption for each food item in nine predetermined categories, ranging from ‘never or less than once per month’ to ‘six or more times per day’. A standard portion size was assigned to each food item.

The daily consumption of each item on the FFQ (in grams per day) was determined by multiplying its intake frequency by the standard portion size. If applicable, a seasonal variation factor was also taken into account (i.e., a factor of 0.25 was considered equivalent to consumption over a 3-month period). The food was then converted into nutrients using the Food Processor Plus software (Version 11.5, ESHA Research, Salem, OR, USA), which utilizes nutritional information from the food composition tables of the US Department of Agriculture. The database has been adapted to reflect typical Portuguese foods.

#### 2.3.4. Mediterranean-Diet Score

Adherence to the Mediterranean diet was calculated based on the FFQ, using the original methodology [21] and the 9-point score proposed by Trichopolou et al. in 2003 [26]. Participants were classified into three levels of adherence to the Mediterranean score: low adherence (0–3 points); medium adherence (4–6 points), and high adherence (7–9 points). A score of 9 indicates that the participant meets all the characteristics of the Mediterranean diet.

### 2.4. Statistical Analysis

Statistical analyses were performed using the IBM Statistical Package for the Social Sciences for Windows^®^ (Version 27.0. IBM Corp, Armonk, NY, USA). A *p*-value < 0.05 was regarded as significant.

For the purpose of this study, participants were divided into tertiles according to their total SHED Index score (1st tertile: ≤55.28; 2nd tertile: 55.29–65.70; 3rd tertile: ≥65.71). The Kolmogorov–Smirnov test was used to assess normality. To compare continuous variables across tertiles of the total SHED Index score, we used a one-way analysis of variance (ANOVA) or the Kruskal–Wallis test, and the Chi-square test was used to test for categorical variables. Person (r) or Spearman (ρ) correlation coefficients were used to test the association between variables. Descriptive analysis is presented in terms of means and standard deviations, median and 25th and 75th percentiles, or absolute and relative frequency.

Internal consistency was measured by Cronbach’s alpha and item–total correlation. The internal consistency of the six-point scale was considered acceptable if equal to or higher than 0.70. Items were considered inconsistent with the others of the same domain if their correlation was lower than 0.20 [27], and/or if Cronbach’s alpha increased after their elimination.

To determine the construct validity of the total SHED Index score, we tested the following hypotheses: (i) the total SHED Index score was inversely correlated with the proportion of animal-source foods to the total food weight intake, and (ii) the mean proportion of animal-source foods to the total food weight intake was lower in the highest tertile of the total SHED Index score. In this study, animal-source foods considered were milk, yogurt, cheese, butter, meat, meat products, fish and seafood, and eggs. The criterion validity was examined by testing the relationship between the total SHED Index score and adherence to the Mediterranean Diet. This diet encompasses all the dimensions considered in a healthy and sustainable pattern, including health and nutrition benefits, low environmental impacts, preservation of biodiversity, and promotion of sociocultural food values and the local economy [28].

Reproducibility was assessed by determining agreement and reliability. The agreement for the total SHED Index score was evaluated using Bland–Altman method [29]. The limits of agreement were calculated using the formula: mean change in scores of repeated measurements ±1.96 × standard deviation of these changes. To assess test–retest reliability, an intra-class correlation coefficient (ICC) based on a two-way mixed model was calculated. An ICC lower than 0.5 indicates poor reliability, between 0.5 and 0.75 indicates moderate reliability, between 0.75 and 0.9 indicates good reliability, and above 0.9 indicates excellent reliability [30].

## 3. Results

Descriptive characteristics of the sample according to SHED Index score tertiles are shown in Table 1. Participants in the highest tertile of the SHED Index score were older, mostly women, mostly married, less likely to have a high animal-based diet, and had higher education levels compared to their counterparts. The median of all the SHED Index sub-scores was significantly higher in the highest tertile of the SHED Index score (*p* < 0.01, for all).

The descriptive statistics for the total SHED Index score and its sub-scores, the percentage of participants scoring the lowest and highest scores, and the correlations of sub-scores with the total SHED Index score are presented in Table 2. For the total SHED Index score and all sub-scores, there was no floor effect (Table 2). A ceiling effect (>15% of participants with higher scores) was present only in the Soda score. The majority of participants reported never or rarely consuming soft drinks (66.3%), diet beverages (64.3%), or energy drinks (91.4%).

The total SHED Index score was strongly correlated with Healthy Eating and Sustainable Eating scores (r = 0.894 and r = 0.793, respectively, *p* < 0.001). The lowest correlation coefficients were seen with BVF and Ready Meal scores (r = 0.285 and r = 0.145, respectively, *p* < 0.001) (Table 2). Fruits and vegetables are more frequently purchased at supermarkets (81%), and never or rarely purchased directly from the farmer (79.3%), at a farm (83.0%), store or green-grocer (81%), or delivered from a supermarket to the home (93.9%). Most of the participants reported eating homemade or home-cooked food (85.3%) and cooking food by themselves (83.3%).

### 3.1. Internal Consistency

Two of the six scores had Cronbach’s alpha coefficients greater than 0.70, indicating good internal consistency, while the other four had very low Cronbach’s alpha coefficients ranging from 0.409 to 0.477 (Appendix A). The Cronbach’s alpha reliability coefficient increased for the BFV Location Score and for the Water Score if item 8 (supermarket—Shop in person) and item 2 (home water filter use), respectively, were removed. The item–total correlations were higher for the Healthy Eating Score (0.373 to 0.591) and the Sustainable Eating Score (0.302 and 0.529). Lower item–total correlations were seen for the BFV Location Score (ranging from 0.015 to 0.380), the Ready-Meals Score (ranging from 0.154 to 0.298), the Water Score (ranging from 0.048 to 0.502), and the Soda Score (ranging from 0.266 to 0.358) (Appendix A).

### 3.2. Validity

The proportion of animal-source foods to the total food weight intake was negatively correlated with the total SHED index score (r = −0.281, *p* < 0.001). Participants in the highest tertile of the total SHED index score had a lower proportion of animal-source foods to the total food weight intake compared to their counterparts (1st tertile: 30.1 ± 14.40; 2nd tertile: 24.3 ± 9.81; 3rd tertile: 21.6 ± 9.37, *p* < 0.001) (Figure 1).

A moderate correlation was found between total SHED index score and adherence to the Mediterranean diet (ρ = 0.406, *p* < 0.001). Almost 86% of participants in the highest tertiles of the total SHED index score had medium to high adherence to the Mediterranean diet. A higher median adherence to the Mediterranean diet was found for participants in the highest tertile of the total SHED index score (1st tertile: 3 (2–5); 2nd tertile: 4 (3–6); 3rd tertile: 5 (4–6) *p* < 0.001). Participants in the third tertile of the total SHED index score had higher consumption of vegetables, legumes, fruits, and nuts, as well as fish, and lower consumption of cereals, meat, and dairy products (*p* < 0.05) (Table 3).

### 3.3. Reproducibility

Figure 2 shows the Bland–Altman plot, where 97.6% of the data points fall within the limits of agreement. The mean difference between the two measurements times was 0.32 ± 6.61 points. No relationship was observed between the error and the mean value.

Good reliability was found for the total SHED index score (ICC = 0.772, 95% confidence interval (CI): 0.669–0.846), the Soda Score (ICC = 0.783, 95% CI: 0.685–0.854), and the Healthy Eating Score (ICC = 0.754, 95% CI: 0.612–0.843). The Water Score (ICC = 0.686, 95% CI: 0.555–0.785) and Sustainable Eating Score (ICC = 0.539, 95% CI: 0.369–0.679) presented moderate reliability, while the BFV Location Score (ICC = 0.378, 95% CI: 0.181–0.546) and the Ready Meals Score (ICC = 0.399, 95% CI: 0.188–0.571) exhibited poor reliability.

## 4. Discussion

The present study describes the adaptation and validation of the SHED Index for the Portuguese adult population. The SHED Index was originally developed by Tepper et al. [21] and validated in an Israeli population aged 20–45 years. It is a simple tool that is quick and easy to apply and understand, designed to assess and rank healthy and sustainable diets. The Index includes elements from different dimensions of food sustainability, such as environmental, sociocultural, economic, and nutritional aspects of individual diets. The results of this study show that the SHED Index is a valid tool for assessing sustainable and healthy diets, providing reliable and reproducible results.

Our results also show that women, older individuals, and those with a high level of education seem to have a more sustainable and healthy diet than their counterparts, as demonstrated in the Israeli population [18] and in other studies [18,31,32]. Contrary to expectations, there were no differences in BMI values between the tertiles of the SHED Index score. Non-overweight individuals seem to have a healthier and more sustainable diet, although this difference was not statistically significant. In contrast to the findings of this study, other authors have shown that individuals with a normal weight have a higher consumption of healthy and sustainable foods, suggesting a potential protective role of sustainable diets in preventing weight gain, becoming overweight, and obesity [33,34,35].

Sustainable and healthy diets involve a complexity of dimensions, and there is no gold standard for assessing them. Thus, some indicators already identified in the literature related to a sustainable dietary pattern were used, namely the proportion of consumption of animal-sourced and vegetable-sourced foods, as well as adherence to the Mediterranean dietary pattern, which has been deemed sustainable by various experts [28,36]. We confirmed construct validity based on established criteria. A higher total SHED Index score was associated with moderate to high adherence to the Mediterranean diet, while it was inversely related to the proportion of animal-sourced foods in the overall food intake.

The Mediterranean diet has been described as a sustainable model, with lower environmental impacts [4,37,38]. This dietary pattern is characterized by a plant-based diet with small to moderate serving sizes based on frugality and local habits. It emphasizes a preference for local, seasonal, fresh, and minimally processed food, supporting biodiversity and eco-friendly and traditional foods [39]. Animal protein sources should be consumed less frequently, on a weekly basis, and should only make up a small portion of total food intake. Animal and sugar-rich processed foods (e.g., processed meat, pastries, and sweets) should only be consumed occasionally [39]. These items were also considered in the SHED index, which measures the sustainability of a diet. A more sustainable diet is associated with a lower proportion of animal-sourced foods compared to vegetable-sourced foods, and with a high consumption of vegetables and fruits, as recommended by the World Health Organization (400 g daily) [40]. The SHED index also positively scores the low consumption of salt, salty products, sugar, sugary products, processed food, and ready meals, which should be avoided according to the Mediterranean pattern due to their high environmental footprint and association with various health conditions [41,42,43]. On the other hand, the SHED Index positively scores practices that reduce food waste, such as reusing leftovers and practicing recycling. It also scores the preference for local and organic food products and homemade meals, which align with the principles of the Mediterranean diet that considers not only the specific foods and nutrients included, but also the way the food is produced, cooked, and consumed [44].

According to Dernini et al. [44], the Mediterranean diet incorporates four important dimensions and sustainable benefits. These include major health and nutrition benefits, low environmental impacts and richness in biodiversity, high sociocultural food values, and positive local economic returns. All of these dimensions align with the definition of sustainable diets [3].

The validity of the SHED Index was also confirmed by the ratio of animal-sourced foods to the total food weight intake. High consumption of animal-sourced foods has been associated with greenhouse gas emissions, environmental harm, and negative health effects, increasing the risk of adverse outcomes from major chronic diseases [31,45,46]. Furthermore, according to the Global Burden of Disease study [47], consumption of red meat emerged as one of the primary factors contributing to years lost due to illness, disability, or premature death.

As expected, participants in the third tertile of the total SHED Index score presented a high consumption of vegetables, legumes, fruits, and nuts. These foods are associated with a sustainable food pattern since they have a low environmental footprint [1]. On the other hand, participants with a high SHED Index score showed a low consumption of cereals, meat, and dairy products. Meat consumption has been associated with several adverse health and environmental effects [31,45,48]. According to Springmann et al. [49], transitioning from animal-sourced foods to a plant-based diet will greatly reduce environmental impact, improve nutritional consumption, and reduce the risk of premature mortality. The EAT LANCET study [4] suggests that shifting towards a dietary pattern that includes more plant-based foods and fewer animal-sourced foods would offer environmental benefits, nutrient adequacy, and improved health.

Concerning internal consistency, four out of the six sub-scores determined had very low Cronbach’s alpha coefficients. In another study that developed and validated the Planetary Healthy Diet Index [16], the authors referred to the low value for Cronbach’s as acceptable since the diet is a complex and multidimensional construct, and the fact that it is applied to a large and heterogeneous population sample affects internal consistency, which is aligned with our findings.

Regarding reproducibility, we found that the total SHED Index score has good agreement and reliability. However, when analyzing the sub-score reliability, the BFV Location Score and the Ready Meals Score presented lower reliability compared to the other sub-scores. Reliability coefficients are expected to be low when there is little variability among the scores, which occurred when most or all participants provided identical ratings [50]. Upon analyzing the data, we observed that the majority of the participants reported never or rarely purchasing fruit and vegetables through direct delivery or from a farmer’ box, directly at a farm, at a country store/green grocery, or through supermarket home delivery. As for ready meals, most participants reported cooking food themselves or taking part in preparing it and eating homemade or home-cooked food daily or almost daily. The SHED Index was developed to assess both healthy dietary patterns and pro-sustainability behaviors, thus lower reliability in these sub-scores does not necessarily diminish its value as a whole but may limit the isolated use of the BFV Location Score and the Ready Meals Score.

Despite other indexes that have been proposed by different authors [17,18,19,20], the total SHED Index score includes items from different dimensions, including food consumption, consumer habits, eating behavior (food purchasing and preparation), food waste, and social aspects, encompassing all the different dimensions of the sustainable diet concept. Other existing indices have presented some disadvantages when compared with the SHED Index. Some of them only consider the consumption of specific food groups, such as the WISH [19] and PHDI [16]. The SDI [18] does not take into account food waste, and the HSDI [17] does not consider the origin and acquisition of food products. Therefore, the SHED Index is an easily understandable and applicable tool, overcoming the limitations of other methodologies, namely the length and time it takes to apply, as well as the absence of a quantitative methodology to rank healthy and sustainable diets.

It is important to highlight that the different dimensions evaluated through the SHED Index are in line with Portuguese nutritional guidelines. In 2016, the Portuguese food guide (Food Wheel Guide) was updated with the aim of emphasizing the characteristics of the Mediterranean food pattern [51]. Currently, the national recommendations advocate for a greater consumption of plant-based foods instead of animal products, as well as a reduced consumption of processed foods, sugary products, and salty products. In addition, there is a focus on respecting seasonality and favoring locally sourced food. The promotion of traditional healthy cooking techniques and the encouragement of time dedicated to cooking are also emphasized, with the goal of transmitting these practices between generations.

Based on our findings, the SHED index score seems to be a good indicator for assessing healthy and sustainable diets. It demonstrates satisfactory results in relation to criteria such as validity and reliability.

This study presents some limitations. Firstly, the SHED Index score was validated based on the food consumption obtained by an FFQ, which has some limitations, namely the memory bias associated with recalling the last year’s consumption through an extensive list of foods. Nonetheless, the FFQ used in this study has been validated for the Portuguese adult population and is widely used in epidemiological studies. Moreover, the purpose of using the FFQ in this study was not to assess actual dietary intake. We utilized the FFQ to rank participants’ consumption. In this case, the relative ranking may be more significant than the precision in the dietary assessment. The FFQ proves useful in classifying individuals based on ranking [52,53,54]. Another limitation is the potential for incorrect estimation of food portion sizes, so a semi-quantitative questionnaire was used to help respondents remember the amount they consumed. Finally, since our sample contained mainly young adults with a female majority, the generalizability may be affected. Further research is needed to confirm these findings in other population groups.

## 5. Conclusions

Our findings suggest that the SHED Index is a valid and reliable tool for assessing sustainable and healthy diets in the Portuguese adult population. This index allows us to rank diets based on both healthy dietary patterns and pro-sustainable behaviors. It can be very useful and valid for intervention studies and epidemiological studies with large sample sizes, such as accessing sustainable nutrition behaviors. Additionally, it can be a valuable tool for healthcare professionals and dietitians’ in their daily practice.

The transformation of dietary patterns towards healthy and sustainable diets is imperative for achieving the United Nations Sustainable Development Goals and the Paris Agreement. In this sense, the existence of validated tools for different countries is an important step towards quantitatively monitoring the sustainability of diets, considering the characteristics of the country’s population and eating habits. Thus, the SHED Index can be a valid tool and an excellent support for achieving a more sustainable food consumption and a more sustainable food system. This, in turn, will help achieve international goals, while also having a positive impact on individuals’ nutritional status and health.

## Figures and Tables

**Figure 1 nutrients-15-05071-f001:**
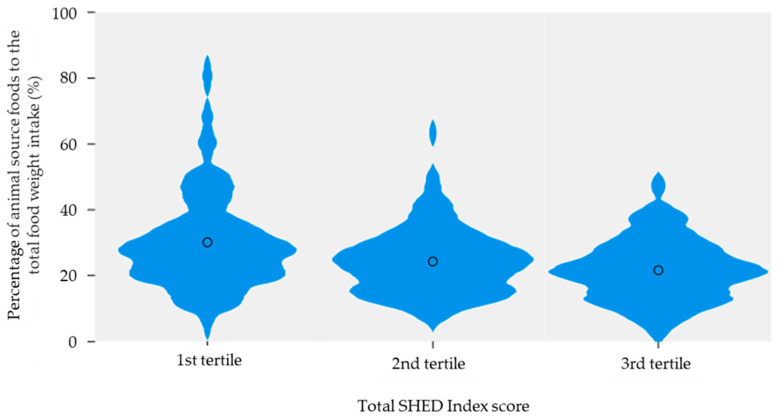
Violin plot showing the distribution of the proportion of animal-source foods to the total food weight intake according to SHED Index score tertiles (black circle represents the mean).

**Figure 2 nutrients-15-05071-f002:**
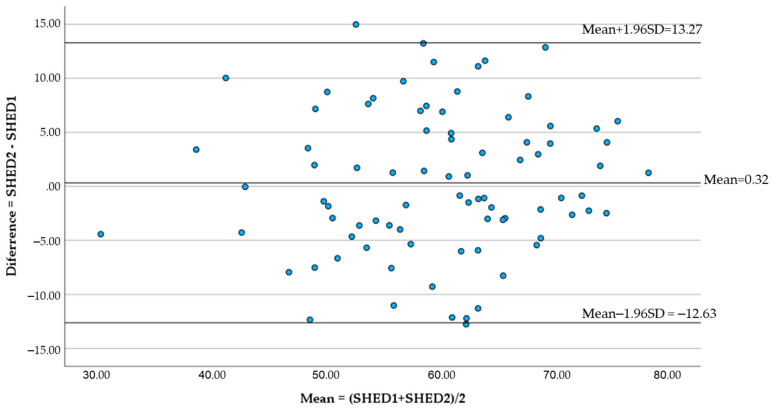
Bland–Altman plot with representation of the limits of agreement and mean differences between the two measurements times. SHED1, total SHED index score in the first measurement; SHED2, total SHED index score in the second measurement.

**Table 1 nutrients-15-05071-t001:** Characteristics of the sample according to the SHED Index score tertiles.

	Total (n = 347)	Tertiles of the SHED Index Score ^a^			*p* ^b^
		1st Tertile (n = 116)	2nd Tertile (n = 117)	3rd Tertile (n = 114)	
Age, years	27.0 (20.0; 50.0)	21.0 (19.0; 31.50)	24.0 (21.0; 48.0)	49.0 (24.0; 56.25)	<0.001
Sex (n, % women)	224 (64.6)	55 (47.4)	80 (68.4)	89 (78.1)	<0.001
Marital Status (n, % married)	128 (36.9)	26 (22.4)	34 (29.1)	68 (59.6)	<0.001
Education level, n (%)					
Mandatory or less	34 (9.8)	10 (8.6)	9 (7.7)	15 (13.2)	0.016
Secondary	190 (54.8)	74 (63.8)	68 (58.1)	48 (42.1)	
College/university	123 (35.4)	32 (27.6)	40 (34.2)	51 (44.7)	
Household (number of individuals)	3.0 (3.0; 4.0)	3.0 (3.0; 4.0)	3.0 (3.0; 4.0)	3.0 (3.0; 4.0)	0.887
Level of urbanization, n (%)					
City	129 (37.2)	48 (41.4)	41 (35.0)	40 (35.1)	0.162
Peripheral city	113 (32.6)	43 (37.1)	37 (31.6)	33 (28.9)	
Village/community settlement	105 (30.3)	25 (21.6)	39 (33.3)	41 (36.0)	
Weight Status, n (%)					
Non-overweight	219 (63.1)	78 (67.2)	73 (62.4)	68 (59.6)	0.589
Overweight	91 (26.2)	25 (21.6)	34 (29.1)	32 (28.1)	
Obese	37 (10.7)	13 (11.2)	10 (8.5)	14 (12.3)	
Dietary pattern, n (%)					
Omnivore	310 (89.3)	102 (87.9)	103 (88.0)	105 (92.1)	0.002
High animal-based diet (paleo + ketogenic)	29 (8.4)	12 (10.3)	14 (12.0)	3 (2.6)	
Vegetarian/Vegan/Plant-based	8 (2.3)	2 (1.7)	0 (0)	6 (5.3)	
Physical Activity (min/week)	300.0 (150.0; 630.0)	337.5 (160.0; 762.75)	300 (149.50; 600.0)	295.0 (147.50; 600.0)	0.377
Energy intake (kcal/day)	2139.4 (1713.56; 2736.43)	2274.6 (1690.20; 3056.65)	2082.19 (1719.42; 2760.47)	2119.5 (1708.65; 2398.26)	0.170
SHED Index score	60.8 (53.24; 67.32)	50.4 (45.86; 53.24)	61.25 (57.95; 63.48)	69.8 (67.40; 73.07)	<0.001
Healthy eating	20.0 (16.0; 25.0)	14.0 (10.0; 17.0)	20.0 (18.0; 23.0)	26.0 (24.0; 28.0)	<0.001
Sustainable eating	12.0 (9.9; 15.0)	8.0 (6.0; 10.0)	12.0 (10.0; 14.0)	16.0 (14.0; 18.0)	<0.001
BFV location	3.3 (2.38; 4.25)	2.7 (1.88; 3.50)	3.5 (2.75; 4.50)	3.6 (2.75; 4.50)	<0.001
Ready meals	2.7 (1.94; 3.33)	2.0 (1.33; 3.0)	2.7 (2.02;3.06)	3.2 (2.50; 3,67)	<0.001
Water	1.3 (0.0; 2.0)	0.8 (0.0; 1.75)	1.3 (−0.13; 2.0)	1.5 (0.50; 1.25)	0.006
Sodas	−1.0 (−2.0; −0.33)	−2.0 (−3.29; −1.0)	−1.0 (−2.0; −0.67)	−0.67 (1.0; 0.0)	<0.001

Data are presented as median (P25; P75) or number (percentage); BFV, Buy Fruits and Vegetables. ^a^ 1st tertile: ≤55.28; 2nd tertile: 55.29–65.70; 3rd tertile: ≥65.71. ^b^ *p* value from Kruskal–Wallis test or Chi-Square test as appropriate.

**Table 2 nutrients-15-05071-t002:** Descriptive statistics (mean, SD, range of score), the percentage of participants scoring the lowest and highest scores, and correlations of sub-scores with the total SHED Index score.

	Mean Score (SD)	Min. Score	Max. Score	Participants with Min. Score (%)	Participants with Max. Score (%)	r ^a,b^
SHED Index score	60.0 (10.0)	32.6	81.1	0.3	0.3	-
Healthy eating	19.7 (6.2)	2.0	30.0	0.9	3.7	0.894
Sustainable eating	11.8 (4.4)	1.0	21.0	1.2	3.2	0.793
BFV location	3.3 (1.3)	0.4	7.4	1.4	0.3	0.285
Ready meals	2.5 (1.0)	−0.1	4.5	0.3	0.3	0.145
Water	1.0 (1.2)	−1.5	3.8	0.6	1.2	0.435
Sodas	−1.5 (1.4)	−7.5	0.0	0.3	20.0	0.534

BFV, Buy Fruits and Vegetables; Max, maximum; Min, minimum; SD, standard deviation; ^a^ Pearson correlation coefficient with the SHED Index score; ^b^ *p*-value < 0.001 for all.

**Table 3 nutrients-15-05071-t003:** Mediterranean-Diet adherence score and dietary intake of Mediterranean Diet components for the total sample, as well as according to tertiles of the SHED Index Score.

	Total (n = 347)	Tertiles of the SHED Index Score ^a^			*p* ^b^
		1st Tertile (n = 116)	2nd Tertile (n = 117)	3rd Tertile (n = 114)	
Mediterranean-Diet adherence score	4.0 (3.0; 6.0)	3.0 (2.0; 5.0)	4.0 (3.0; 6.0)	5.0 (4.0; 6.0)	<0.001
Mediterranean-diet adherence (n, %)					
Low	125 (36.0)	71 (61.2)	38 (32.5)	16 (14.0)	<0.001
Medium	121 (34.9)	25 (21.6)	45 (38.5)	51 (44.7)	
High	101 (29.1)	20 (17.2)	34 (29.1)	47 (41.2)	
Mediterranean diet components					
Vegetables (g/day)	154.7 (88.28; 251.04)	103.7 (44.76; 178.65)	152.3 (103.02; 280.32)	195.6 (144.31; 277.54)	<0.001
Legumes (g/day)	37.5 (17.51; 88.93)	28.5 (17.51; 78.18)	37.5 (23.79; 112.51)	62.9 (25.70; 112.51)	<0.001
Fruits and nuts (g/day)	308.8 (185.97; 465.58)	211.4 (110.32; 355.93)	317.3 (206.49; 484.07)	383.1 (261.83; 508.59)	<0.001
Cereals (g/day)	262.6 (184.95; 402.67)	318.7 (187.82; 443.53)	268.1 (198.66; 401.90)	226.4 (179.16; 336.36)	0.012
Fish (g/day)	74.1 (42.29; 136.10)	60.4 (34.92; 118.60)	82.1 (44.54; 150.38)	83.1 (49.81; 140.54)	0.009
MUFA/SFA ratio	1.6 (1.37; 1.97)	1.4 (1.25; 1.62)	1.6 (1.40; 1.92)	1.9 (1.58; 2.25)	<0.001
Meat (g/day)	120.0 (80.00; 157.63)	127.4 (95.7; 181.34)	121.3 (80.00; 156.63)	107.0 (69.62; 126.19)	<0.001
Dairy products (g/day)	125.0 (41.42; 246.00)	137.2 (57.86; 307.46)	121.8 (44.44; 223.02)	83.1 (28.03; 171.43)	0.017
Alcohol (g/day)	1.7 (0; 5.79)	1.9 (0; 6.73)	1.6 (0; 5.64)	1.7 (0; 5.84)	0.683

Data are presented as median (P25; P75) or number (percentage); MUFA, Monounsaturated fatty acids; SFA, Saturated fatty acids. ^a^ 1st tertile: ≤55.28; 2nd tertile: 55.29–65.70; 3rd tertile: ≥65.71. ^b^ *p* value from Kruskal–Wallis test or Chi-Square test as appropriate.

## Data Availability

Data are contained within the article and Appendix A.

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
