# Peer review of "The SHED Index: A Validation Study to Assess Sustainable HEalthy Diets in Portugal"

_nutrients, 2023, doi:10.3390/nu15245071_

Round 1
Reviewer 1 Report
Comments and Suggestions for Authors
The study discusses the adaptation and validation of the SHED Index for assessing sustainable and healthy diets in the Portuguese adult population. The SHED Index was originally developed for the Israeli population and considers various dimensions such as environmental, sociocultural, economic, and nutritional aspects. The study emphasizes the importance of validated tools like the SHED Index in monitoring and promoting sustainable and healthy diets, aligned with international goals like the UN's Sustainable Development Goals and the Paris Agreement.
As a reviewer, I find the study to be generally well-conducted with a focus on an important topic: assessing sustainable and healthy diets using the SHED Index. However, there are some points that could be further clarified or expanded upon for a more robust understanding:
- Limitations Acknowledgment: While the study mentions limitations like the FFQ's memory bias, it might be beneficial to discuss how these limitations could have impacted the results.
- Sub-score Reliability: The low reliability of some sub-scores should be addressed. Could a refinement of these sub-scores improve the index's overall reliability?
- Generalizability: The sample seems skewed towards younger and female individuals. How does this affect the applicability of the SHED Index across diverse demographics?
- Comparison with Other Tools: The study could benefit from a comparison with other existing indices to highlight the unique advantages or disadvantages of the SHED Index.
- Practical Implications: While the study concludes that the SHED Index can be useful for intervention studies, some concrete examples or case studies would provide a practical context.
- Nutritional Aspects: Given the multi-dimensionality of the SHED Index, a more detailed discussion on how it aligns or contrasts with nutritional guidelines could be beneficial.
English is ok
Author Response
The reviewer’s comments are outlined below with our point-by-point responses.
We sincerely thank the reviewer for their thoughtful and generous comments regarding our submitted manuscript. We greatly appreciate the time and effort invested in reviewing our work and providing such positive feedback. We are grateful for the specific points the reviewer highlighted, which have enabled us to improve our manuscript.
Taking into account the reviewer’s suggestions, all the changes made in the original manuscript are presented using the "Track Changes" function in Microsoft Word.
The reviewer’s comments are outlined below with our point-by-point responses
1. Limitations Acknowledgment: While the study mentions limitations like the FFQ's memory bias, it might be beneficial to discuss how these limitations could have impacted the results.
Authors’ reply: In our opinion, this bias is not differential, since we do not consider the consumption of nutrients as the primary goal. Instead, we t rank the participants, relative to each other, so this is not an issue. Nevertheless, in response to the reviewer’s remark, we have made improvements to the discussion.
2. Sub-score Reliability: The low reliability of some sub-scores should be addressed. Could a refinement of these sub-scores improve the index's overall reliability?
Authors’ reply: The authors appreciate the reviewer’s comment, and we have added information on this point in the discussion section.
3. Generalizability: The sample seems skewed towards younger and female individuals. How does this affect the applicability of the SHED Index across diverse demographics?
Authors’ reply: In the authors’ opinion, it does not affect the applicability of the SHED Index. It is important to point out that our sample included 35.4% men (n=123), which cannot be overlooked, considering that the Portuguese population has more women than men. Additionally, as we discussed, our results are in line with other studies that have shown that women have a more sustainable diet than men. This point has been adressed in the discussion section.
4. Comparison with Other Tools: The study could benefit from a comparison with other existing indices to highlight the unique advantages or disadvantages of the SHED Index.
Authors’ reply: Considering the suggestion from the reviewer, we have also included information oabout how the SHED Index compares to others in the discussion section.
5. Practical Implications: While the study concludes that the SHED Index can be useful for intervention studies, some concrete examples or case studies would provide a practical context.
Authors’ reply: Considering the suggestions provided by the reviewer, we have added concrete examples to the conclusions section.
6. Nutritional Aspects: Given the multi-dimensionality of the SHED Index, a more detailed discussion on how it aligns or contrasts with nutritional guidelines could be beneficial.
Authors’ reply: In response to reviewer suggestions, we have included a more detailed discussion on how the multi-dimensionality of the SHED Index aligns with Portuguese Nutritional guidelines.
Reviewer 2 Report
Comments and Suggestions for Authors
Review of the article titled “The SHED Index: a validation study to assess Sustainable 2 HEalthy Diets in Portugal”
Major comments: The reviewed manuscript concerns the currently widely researched both direct and indirect impact of diet quality on human health through the impact on the environment of his life. The introduction is interestingly written and contains a comprehensive justification for the development of a new indicator describing the quality of the diet. Selection of methods and tools adequate to the assumed goal. It is worth emphasizing the very careful description of the research methodology. The statistical analysis and presentation of the results are not objectionable. However, in the reviewer's opinion, the discussion should be supplemented with a comment regarding the lack of differences in BMI values ​​between quartile groups. Since SHED Index is supposed to reflect the impact of diet on human health, such a comment seems necessary. How can we explain the lack of differences in BMI between subjects following a more and less healthy and sustainable diet? Are people with BMI<18.5 were excluded from the study and if so, why? In my opinion, the article is needed to complete the discussion and make the minor corrections indicated below.
Minor comments:
Line 48: should be ‘towards’ instead of ‘to wards”
Table 1. – it is not known what it means ‘nº’ in the line labelled ‘Household (nº person)’
Table 1. – it is not known what the numbers in the line labelled ‘Level of urbanization, n (%)’ mean
Author Response
We sincerely thank the reviewer for their thoughtful and generous comments regarding our submitted manuscript. We greatly appreciate the time and effort invested in reviewing our work and providing such positive feedback. We are grateful for the specific points the reviewer highlighted, which have enabled us to improve our manuscript.
Taking into account the reviewer’s suggestions, all the changes made in the original manuscript are presented using the "Track Changes" function in Microsoft Word.
The reviewer’s comments are outlined below with our point-by-point responses.
Major comments: The reviewed manuscript concerns the currently widely researched both direct and indirect impact of diet quality on human health through the impact on the environment of his life. The introduction is interestingly written and contains a comprehensive justification for the development of a new indicator describing the quality of the diet. Selection of methods and tools adequate to the assumed goal. It is worth emphasizing the very careful description of the research methodology. The statistical analysis and presentation of the results are not objectionable. However, in the reviewer's opinion, the discussion should be supplemented with a comment regarding the lack of differences in BMI values ​​between quartile groups. Since SHED Index is supposed to reflect the impact of diet on human health, such a comment seems necessary. How can we explain the lack of differences in BMI between subjects following a more and less healthy and sustainable diet? Are people with BMI<18.5 were excluded from the study and if so, why? In my opinion, the article is needed to complete the discussion and make the minor corrections indicated below.
Authors’ reply: According to the reviewer's suggestion, the discussion was supplemented with a comment regarding the lack of differences in BMI values ​​between tertile groups. Considering the reduced number of individuals (n=5) classified as underweight and the main purpose of the study, we decided to combine this group with the individuals classified as normal weight. The body mass index cut-points used were presented in the methodology section, under Sociodemographic and lifestyle information.
Minor comments:
Line 48: should be ‘towards’ instead of ‘to wards”
Authors’ reply: The text was rectified.
Table 1. – it is not known what it means ‘nº’ in the line labelled ‘Household (nº person)’
Authors’ reply: It refers to the number of individuals, as clarified in Table 1.
Table 1. – it is not known what the numbers in the line labelled ‘Level of urbanization, n (%)’ mean
Authors’ reply: It was an oversight, so the information has been deleted. The structure of the table has been adapted to make it easier to read and interpret.
Round 2
Reviewer 1 Report
Comments and Suggestions for Authors
The authors improved the paper following my suggestions
Author Response
Thank you very much.
Reviewer 2 Report
Comments and Suggestions for Authors
I accept the revised version of the article.
Author Response
Thank you very much